# Different computations over the same inputs produce selective behavior in algorithmic brain networks

**Katarzyna Jaworska[1], Yuening Yan[1], Nicola J van Rijsbergen[2], Robin AA Ince[1], Philippe G Schyns[1]\***

[1]School of Psychology and Neuroscience, University of Glasgow, Glasgow, United Kingdom; [2]Department of Psychology, Edge Hill University, Ormskirk, United Kingdom

**Abstract** A key challenge in neuroimaging remains to understand where, when, and now particularly *how* human brain networks compute over sensory inputs to achieve behavior. To study such dynamic algorithms from mass neural signals, we recorded the magnetoencephalographic (MEG) activity of participants who resolved the classic XOR, OR, and AND functions as overt behavioral tasks (N = 10 participants/task, N-of-1 replications). Each function requires a different computation over the same inputs to produce the task-specific behavioral outputs. In each task, we found that source-localized MEG activity progresses through four computational stages identified within individual participants: (1) initial contralateral representation of each visual input in occipital cortex, (2) a joint linearly combined representation of both inputs in midline occipital cortex and right fusiform gyrus, followed by (3) nonlinear task-dependent input integration in temporal-parietal cortex, and finally (4) behavioral response representation in postcentral gyrus. We demonstrate the specific dynamics of each computation at the level of individual sources. The spatiotemporal patterns of the first two computations are similar across the three tasks; the last two computations are task specific. Our results therefore reveal where, when, and how dynamic network algorithms perform different computations over the same inputs to produce different behaviors.

**\*For correspondence:**
philippe.schyns@glasgow.ac.uk

**Competing interest:** The authors declare that no competing interests exist.

## Editor's evaluation

How does the brain implement basic logical computations (AND, OR, XOR) regardless of stimulus types? This is one of the most fundamental questions in cognitive neuroscience. This MEG study, by combining interesting experimental paradigms and sophisticated signal analyses, demonstrates four serial neural components in different brain regions that correspond to four system-level computations, respectively.

## Introduction

Extensive studies revealed that the primate visual system comprises the ventral and dorsal pathways, with specific anatomical and functional hierarchical organization (*Van Essen et al., 1992*; *Kravitz et al., 2013*). These pathways compute over the high-dimensional visual input, starting separately in each hemisphere with contralateral detection of simple, small features with small receptive fields, that are then hierarchically integrated into more complex, broader receptive field features (*Bugatus et al., 2017*; *Hubel and Wiesel, 1962*; *Kay et al., 2015*), leading to the integrated face, object, and scene features (*DiCarlo and Cox, 2007*; *Grill-Spector and Weiner, 2014*; *Kriegeskorte et al., 2008*; *Sigala and Logothetis, 2002*) that are compared with memory to produce behavior (*Zhan et al., 2019b*;

*Wyart et al., 2012*; *Ratcliff et al., 2009*; *Alamia and VanRullen, 2019*). This flow of information reverses when the same pathways predict the top-down input from memory (*Friston, 2008*; *Linde-Domingo et al., 2019*; *Engel et al., 2001*).

There is broad agreement that such a bidirectional hierarchical architecture supports much of the information processing that subtends everyday face, object, and scene recognition. However, despite considerable progress, we have yet to understand where, when, and how specific algorithmic computations in the pathways dynamically represent and transform the visual input into integrated features to produce behavior (*Kriegeskorte and Douglas, 2018*; *Naselaris et al., 2011*; *Wu et al., 2006*) and vice versa, when reversing the flow in the hierarchy, to predict a cascade of features from complex to simpler ones. Furthermore, it is unclear that such an algorithmic understanding can be achieved with current analytical approaches to neuroimaging, even in simple tasks (*Jonas and Kording, 2017*). Here, we achieved such systems-level algorithmic understanding with magnetoencephalographic (MEG) measurements, in the context of well-defined visual inputs and tasks.

We framed this broad problem using the classic logical functions XOR, AND, and OR, in which different algorithms are required to produce correct responses from the same input stimuli (see these input-output relationships in *Figures 1 and 2*). XOR is famously a nonlinearly separable function, whereas AND or OR is linearly separable, implying nonlinear vs. linear transformations of the same inputs in the considered architectures (*Minsky and Papert, 2017*; *Rumelhart et al., 1986*; *Gidon et al., 2020*) (see *Figures 1A and 2*). We aimed to reverse engineer the different stages of linear and nonlinear computations in brain networks that implement the algorithms (*O'Reilly and Mars, 2011*).

To do so, we simultaneously presented the inputs laterally within the visual field (a pair of sunglasses on a face, with dark 'on' vs. clear 'off' lenses representing the binary inputs, *Figure 1A*) so that occipital cortex initially represented each separately and contralaterally (i.e. in analogy to a network model that takes in two distinct inputs, see *Figure 1A*, here with left vs. right input projected in right vs. left occipital cortex; see *Methods, Stimuli*). Our analyses of the ensuing computations in the networks of individual participants systematically revealed four distinct stages that represent and transform the same inputs to produce different task-specific behavior. The first two stages of linear computations similarly represent the two inputs across the XOR, AND, and OR tasks. The last two stages of nonlinear computations differently represent the same inputs in a task-dependent manner (N = 10 participants per task, each analyzed separately to provide an independent replication; we further replicated the key results in different participants, using opposite-phase Gabor patches and also with sequentially presented inputs) (*Little and Smith, 2018*; *Ince et al., 2021*; *Naselaris et al., 2021*; *Smith and Little, 2018*). A video (*Figure 1—video 1*) visualizes the key results in one typical participant, with four stages of computation in the brain between stimulus and behavior schematized in a network model, and examples of the different color-coded source-level dynamic computations on the same inputs that comprise each stage. We advise watching (*Figure 1—video 1*) stage by stage, to complement the presentation of the main results below.

## Results

Starting with behavior, *Table 1* shows that participants were both accurate and fast in all tasks, with no significant task differences on average accuracy and reaction times (RT), measured with independent sample t-tests. We reconstructed the dynamic neural representation of the inputs of each participant from concurrently measured, source modeled MEG activity (see *Methods, MEG Data Acquisition, Source Reconstruction*).

To simplify presentation, henceforth we use vector notation to denote the state of the two inputs and, for example, write left input 'on', right input 'off' as [1,0]. To preview the analysis and key results, for each source and every 4ms we fit linear models to explain the 2D MEG magnetic vector field activity in terms of the two presented binary inputs, with and without a nonlinear interaction term between them. The interaction term captures the nonlinear integration of the two inputs on this MEG source and time point—i.e., when source response to [1,1] differs from the sum of the responses to [1,0] and [0,1]. Additional metrics quantified how the 2D MEG responses match the response pattern expected in each task (see *Methods, Representational Patterns*). Our analyses reveal that individual MEG source responses reflect changing representations of the visual inputs in the brain, revealing four different stages of neural computations that lead to behavior in each task (see *Methods, Linear vs. Nonlinear Representations, Representation Patterns* and *Figure 1—video 1*).

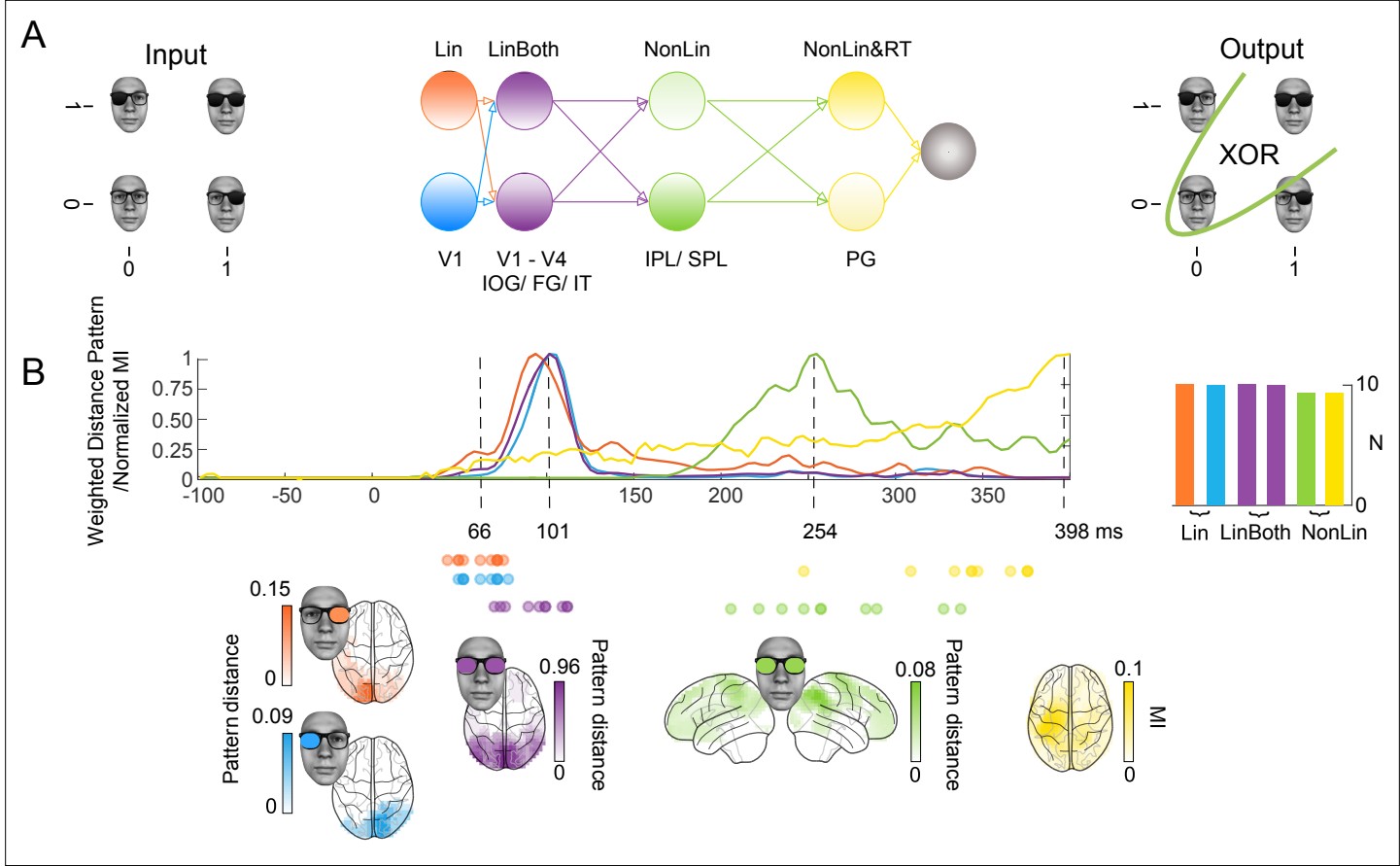

**Figure 1.** XOR task, four stages of computation in a schematic network and in the brain. (**A**) Hypotheses and schematic hierarchical brain network in the XOR task. Stimuli consisted of the image of a face wearing glasses, with dark ('1', 'on') and clear ('0', 'off') left and right lenses serving as inputs, for a total of four input classes for XOR behavioral decisions. (**B**) Four hierarchical stages of computations. Each colored curve shows the average (N = 10 participants) time course of the maximum across sources that: (1) linearly discriminates in its magnetoencephalographic (MEG) activity the 'on' vs. 'off' state of the left (Lin, blue) and right (Lin, orange) inputs (weighted distance pattern), (2) linearly discriminates both inputs (LinBoth, magenta) (weighted distance pattern), (3) nonlinearly integrates both inputs with the XOR task pattern (NonLin, green) (weighted XOR pattern metric) and (4) nonlinearly integrates both inputs with the XOR task pattern and with amplitude variations that relate to reaction time (RT) (mutual information, MI (MEG; RT)) (yellow). Colored brains localize the regions where these computations start (onset times for left and right) or peak (peak latencies for both, XOR and RT) (p<0.05) familywise error rate corrected with a permutation test, (see *Methods, Linear vs. Nonlinear Representations; Representation Patterns*). Dots report the onset time of computation (1) and (2) and the peak time of computation (3) and (4) in each participant. See *Table 2* and *Figure 1—figure supplement 2* for individual participant replications of each computation in the same brain regions and time windows.

The online version of this article includes the following video and figure supplement(s) for figure 1:

**Figure supplement 1.** Clustering of computation stages.

**Figure supplement 2.** Results for individual participants.

**Figure supplement 3.** Replication of four stages of computation with Gabor stimuli.

**Figure 1—video 1.** Dynamic summary of the four stages of computation in the XOR task (data from the example XOR participant highlighted in Figure 1—figure supplement 2).

https://elifesciences.org/articles/73651/figures#fig1video1

## Four systems-level stages of computation link stimulus to behavior

First, we performed a data-driven clustering analysis that delivered four stages of computation in the XOR, AND, and OR tasks (see *Methods, Clustering the Stages of Computation* and *Figure 1—figure supplement 1*). *Figure 1B* (XOR) and *Figure 2* (AND and OR) show the time course of these four stages averaged across the 10 participants of each task (*Figure 1—figure supplement 2* shows the results of each individual participant). Each task shows a similar dynamic unfolding: the first two stages represent and linearly discriminate the visual inputs; the third and fourth stages nonlinearly integrate

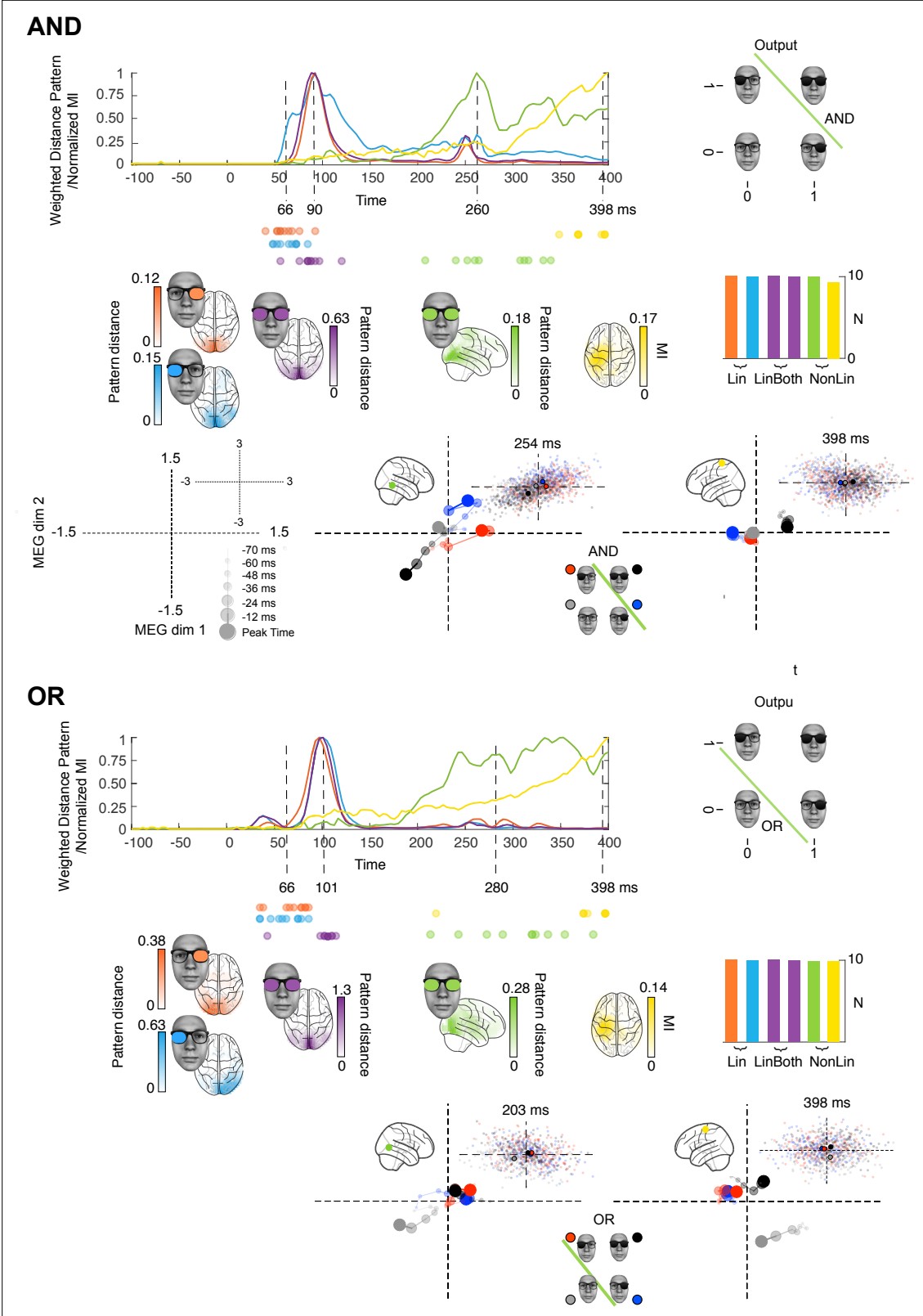

**Figure 2.** Stages of computations in AND and OR tasks. Each colored curve averages (N = 10 participants) the time courses of the maximum across sources that: (1) linearly discriminates (weighted distance pattern) in its magnetoencephalographic activity the 'on' vs. 'off' state of the left (blue) and right (orange) inputs, (2) linearly discriminates both inputs (magenta), (3) nonlinearly integrates both inputs with the respective task pattern and (4) nonlinearly integrates both inputs with the respective task pattern and with amplitude variations that relates to reaction time (RT) (yellow). Colored

*Figure 2 continued on next page*

*Figure 2 continued*

brains localize the regions where these computations start (onset times for left and right) or peak (peak latencies for LinBoth, NonLin, and RT) p<0.05 familywise error rates corrected with a permutation test, see *Methods, Linear vs. Nonlinear Representations; Representation Patterns*. Dots report the onset time of computation (1) and (2), and the peak time of computation (3) and (4) in each participant. See *Table 2* and *Figure 1—figure supplement 2* for individual participant replication of each computation in the same brain regions and time windows. Single source plots in each task develop the data of one typical observer, where the green and yellow computations (cf. color-coded sources in the glass brain) differ between AND, OR, and XOR (see *Figure 3* for caption).

The online version of this article includes the following figure supplement(s) for figure 2:

**Figure supplement 1.** Four stages of computations replicated with delayed stimulus presentation.

**Figure supplement 2.** Four stages of computations replicated with delayed stimulus presentation: individual participants' results.

them in a task-specific manner, revealing the solution of each task in the responses of individual MEG sources. The network model of *Figure 1A* schematizes these stages. Specifically, we show:

1. **Linear, contralateral discrimination of each input state separately ('Lin') in V1-4 regions with onset from ~60ms poststimulus**, quantified as the product of and the LEFT/RIGHT representation pattern metric (see *Methods, Representation Patterns*) and the multivariate $R^2$ of a linear model for each binary input (see *Methods, Linear Representation*), color coded in light blue for the left input, in orange for the right input.
2. **Linear discrimination of both inputs ('LinBoth') on the same occipital and ventral sources ~ 100ms poststimulus**, quantified as the product of the BOTH representation pattern metric (see *Methods, Representation Patterns*) and the multivariate $R^2$ of a linear model considering both inputs with no interaction (see *Methods, Linear Representation*), color coded in magenta.
3. **Nonlinear integration of both inputs ('NonLin') for task performance (XOR, AND, or OR) in temporal-parietal regions ~260ms**, quantified as the product of the XOR representation pattern metric (see *Methods, Representation Patterns*) and the significant improvement in model fit with interaction term (see *Methods, Nonlinear Representation*), color coded in green.
4. **Nonlinear integration of both inputs together with response-related activity ('NonLin&RT') in postcentral gyrus ~400ms**, quantified as mutual information (MI) between the 2D MEG magnetic field and RT on the corresponding trial (see *Methods, Information Theoretic Analyses*), also thresholded by the product of the XOR pattern metric and the model interaction term, color coded in yellow.

In *Figures 1 and 2*, colored sources shown in glass brain localize the regions where each color-coded computation onsets or peaks (cf. dashed lines) in different time windows poststimulus. *Table 2* shows independent replications of each computation within these regions and time windows, in at least 9/10 participants. We also replicated each color-coded computation at the level of individual participants, using Gabor inputs in XOR (*Figure 1—figure supplement 3*) and a sequential presentation of the inputs in XOR, AND, and OR (*Figure 2—figure supplements 1 and 2*). Finally, we show that the four stages of computations

**Table 1.** Mean behavioral accuracy and median reaction times in XOR, AND, and OR, 95% percentile bootstrap confidence interval shown in brackets.
All pairwise comparisons, p>0.05.

|  | Accuracy | Reaction time |
|---|---|---|
| XOR | 98.5% [97, 99] | 499ms [461, 557] |
| AND | 99% [98.6, 99.5] | 457ms [393, 505] |
| OR | 99.3% [99.2, 99.5] | 490ms [439, 541] |

**Table 2.** Number of individual participant replications of the four color-coded computations, within the same region and time window.
Lin, left and right occipital sources [74–117ms]; LinBoth, occipital midline and right fusiform gyrus [74–117ms]; NonLin, XOR: parietal [261–273ms], AND: temporal-parietal [246–304ms], OR: temporal-parietal [261–304ms]; RT, postcentral gyrus [386–398ms]. Bayesian population prevalence (*Ince et al., 2021*) of 9/10 = 0.89 [0.61 0.99]; 10/10 = 1 [0.75 1] (MAP [95% HPDI]).

|  | Lin | LinBoth | NonLin | NonLin&RT |
|---|---|---|---|---|
| XOR | 10/10 | 10/10 | 9/10 (parietal) | 9/10 |
| AND | 10/10 | 10/10 | 10/10 | 9/10 |
| OR | 10/10 | 10/10 | 10/10 | 10/10 |

generalize stage by stage in the XOR from face stimuli to Gabor stimuli (see *Figure 1—figure supplement 1*).

## Detailed dynamic unfolding of each computation at single source level

We next detail the dynamic unfolding of each color-coded computation on single sources, using an exemplar XOR participant (highlighted with colored curves in *Figure 1—figure supplement 2*, XOR, also reported in *Figure 1—video 1*). The selected sources maximize the metric of each computation—i.e., Lin: onset; LinBoth, NonLin, NonLin&RT: peak. The glass brain in *Figure 3A* locates the selected sources and color codes them by type of computation. The subpanels visualize the dynamic response trajectories of each source to the same four stimuli (representing the 'on' vs. 'off' combinatorics of the two inputs) over 72ms, with a 12ms timestep resolution (those indicated with triangle markers in *Figure 1—figure supplement 2* and *Figure 1—video 1*). To preview the results, different source response trajectories to the same inputs detail the neural implementation of the different color-coded computations.

To illustrate, we start with the light blue right occipital source (*Figure 3A*). Its Lin computation (see Legend and *Figure 3B*) develops over seven time-steps (from 2 to 74ms) to linearly discriminate the 'on' (dark) vs. 'off' (clear) state of the left input (see schematic in the bottom left quadrant). The plot shows how 2D source activity progressively separates left lens 'on' trials (red and gray stimuli, see blue line in schematic) from left lens 'off' trials (blue and black stimuli). The adjacent scatter (upper right quadrant) shows the source response to individual trials at 74ms (final plotted time point). The vector diagram (lower right quadrant) confirms that the linear addition of the vector responses [0, 1] and [1, 0] (gray point, sum of blue and red vector) is close to the actual source response to [1, 1] (black point). The left occipital source (orange, *Figure 3A,C*) reflects a similar unfolding for the linear discrimination of the right input 'on' vs. 'off' state. Again, the response is close to the linear sum of the responses to each individual input.

The second computation (LinBoth, magenta) that linearly and jointly represents the 'on' vs. 'off' state of both inputs takes two distinct forms. In midline occipital sources (*Figure 3A,D*), all four stimuli are equally spread out in the quadrants of the source response space (i.e. all inputs are equally discriminated). In contrast, the right fusiform gyrus source (*Figure 3A,E*) discriminates the [1,0] and [0,1] stimuli with an opposite activity, whereas the [1,1] (black dot) and [0,0] (gray dot) stimuli are less discriminated. The vector diagrams of the two LinBoth examples confirm that the joint response to [1,1] is indeed the sum of [1,0] and [0,1] responses. Interestingly, the two LinBoth discriminations illustrate a progression toward an XOR representation. The first LinBoth midline occipital source (*Figure 3D*) discriminates equally each input in the quadrants of its 2D response. In contrast, the amplitude response of the LinBoth right fusiform gyrus source (*Figure 3E*) can linearly discriminate the XOR responses, but only if a nonlinear operation was added (i.e. drawing a circle that separates the two 'same' black and gray stimuli near the origin in the 2D source response space from the two 'different' blue and red stimuli). So, the right fusiform gyrus LinBoth stage likely represents an important intermediate step toward the computation of XOR. We will see next that the green computation adds the nonlinear computation.

The third and most critical computation that starts distinguishing the task (NonLin, green) occurs when the parietal source (*Figure 3A, F*) nonlinearly represents the XOR solution for behavior, with 'same' vs. 'different' stimuli discriminated at 254ms. Like LinBoth in right fusiform gyrus, this representation has black dot [1,1] and gray dot [0,0] responses close together, but with two crucial differences. First, the red and blue vectors (lower right quadrant) now point in the same direction, rather than in opposite directions, as happens in right Fusiform Gyrus LinBoth. Such different source-level responses to the same [1,0] and [0,1] stimuli likely reflect different activities of the neural populations in the regions where the two sources are located. In parietal source NonLin, responses to [1,0] and [0,1] stimuli are magnetic fields with the same dipole orientation, suggesting that the same pattern of cortical activity (i.e. the same neural population) responds to both stimuli. In right fusiform gyrus source LinBoth (*Figure 3A, E*), responses to [1,0] and [0,1] are magnetic fields with different dipole directions, suggesting that different neural populations within the region modeled by the source respond to each stimulus. Second, the representation of [1,1] (black dot) is now nonlinear (green vector), pointing away from the sum of the red and blue vectors of the individual inputs. Following this nonlinear transformation, the XOR outputs are now linearly decodable in the 2D MEG response space.

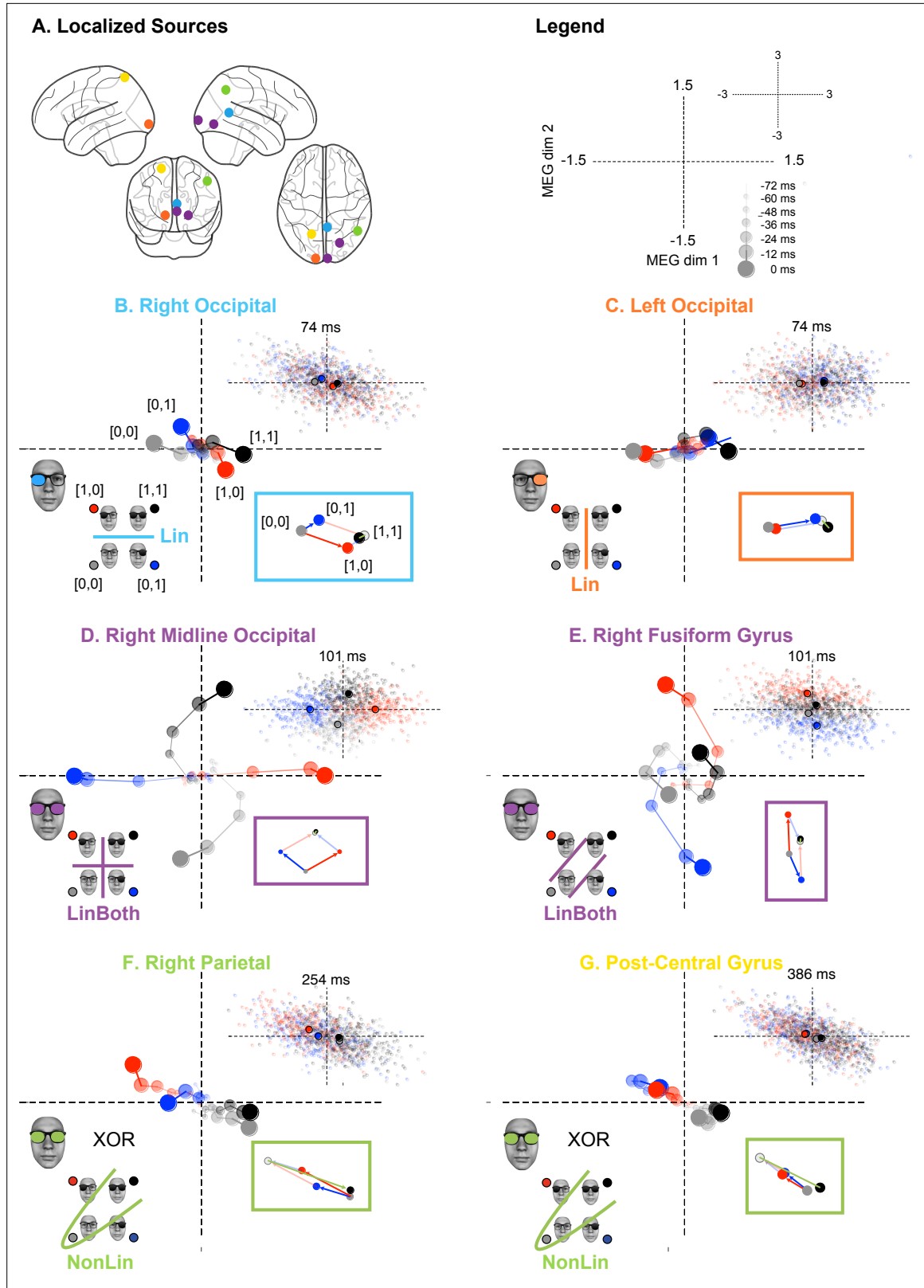

**Figure 3.** Dynamic unfolding of different computations on the same inputs (e.g. XOR participant, see also *Figure 1—video 1* ). (**A**) Localized sources. Color-coded source localized in the top glass brain illustrates each color-coded computations in each scatter plot. Legend. Axes of the scatter plots represent the 2D source magnetic field response. Small dots are single-trial source responses; larger dots their averages for each color-coded stimulus class, dynamically reported over seven timesteps (cf. legend) corresponding to the seven triangular markers in *Figure 1—figure supplement 2*,

*Figure 3 continued*

XOR. Increasing dot sizes, saturations, and connecting lines denote increasing timesteps of the dynamic trajectory. (**B**) Right occipital. The light blue discrimination line indicates the linear (LinLeft) computation that this source represents at the seventh timestep (cf. adjacent scatter for the distribution of individual trials at this time). Inset vector diagram provides a geometric interpretation of the linear computations. Using stimulus [0,0] as the origin: blue arrow illustrates source response to stimulus [0,1] (blue disk); red arrow shows source responses to [1,0] (red disk); gray disk illustrates linear summation of these vectors (opaque lines); black disk is the observed mean response to stimulus [1,1]. (**C**) Left occipital. Same caption for orange LinRight computation. (**D**) Right midline occipital and (**E**) right fusiform gyrus. Same caption for magenta LinBoth computation. (**F**) Right parietal and (**G**) postcentral gyrus. Same caption for green XOR nonlinear computations. Green vector shows discrepant nonlinear observed response to stimulus [1,1] and linear sum of responses to [0,1] and [1,0].

The online version of this article includes the following figure supplement(s) for figure 3:

**Figure supplement 1.** Dynamics of relation of magnetoencephalographic (MEG) responses to reaction times (RTs).

Finally, the fourth stage on a postcentral gyrus source (*Figure 3A, G*, NonLin&RT, yellow) also nonlinearly represents the stimuli, also allowing linear readout of the XOR task outputs at 386ms. In addition, this source activity now relates trial by trial to behavioral RTs. *Figure 3—figure supplement 1* shows that this last postcentral gyrus fourth stage (also in frontal regions) primarily relates to behavioral RTs.

*Figure 2* shows that the key differences in the AND and OR tasks are at the third and fourth stages, where the temporal-parietal (green) and postcentral gyrus (yellow) sources represent AND and OR solutions for task behavior. The earlier stages linearly represent the two inputs, with light blue and orange Lin, magenta LinBoth discriminating the four stimuli as in XOR participants (see *Figure 1—figure supplement 2* for individual replications, prevalence = 1 [0.75 1], MAP [96% HPDI], *Ince et al., 2021*). In NonLin and NonLin&RT stages, the representation is nonlinear and reflects the task (cf. vector diagrams inset, bottom right quadrant). In AND, the task output is linearly separable in the 2D MEG response space: the black [1,1] response is further from the other stimulus classes than they are from each other, see *Methods, Representational Patterns*. In OR, the task outputs are also linearly separable, with the gray [0,0] stimulus class represented apart from the other ones. This shows how these later post-200ms computational stages involve nonlinear task-specific stimulus representations in ventral (NonLin) and parietal (NonLin, NonLin&RT) areas.

## Discussion

Here, we addressed the challenge of understanding where, when, and how the brain dynamically implements different algorithmic computations over sensory inputs. We tightly controlled behavior using the simple logical functions XOR, OR, and AND as tasks that require different computations over the same tightly controlled binary inputs. Our analyses revealed, at the level of individual MEG sources, four main stages of computation that dynamically unfold from ~60 to 400 ms poststimulus. The first computation linearly discriminates the 'on' vs. 'off' state of each input in contral-lateral occipital cortex ~60ms poststimulus. This is followed by the linear discrimination of both inputs on occipital and ventral sources ~100ms, followed by the nonlinear integration of both inputs, revealing the XOR, AND, or OR task solution in 2D source response space in the parietal-temporal regions ~260ms, and finally the nonlinear integration with RT-related activity in postcentral gyrus ~400ms. These four stages are common to XOR, AND, and OR, with the main task-related changes occurring in the latter two nonlinear stages. Notably, we performed all statistical analyses leading to these results within each individual participant, controlling the familywise error rate (FWER) over all considered sources and time points. By treating each participant as an N-of-1 study, 10 participants per task provide 10 independent replications of the experiment. We replicated the four computational stages in at least 9/10 participants (and in two further replication experiments with similarly high prevalence), providing strong evidence that a majority of individuals in the population sampled and tested in the same way would show the same effects (*Ince et al., 2021*).

### Reverse engineering systems-level algorithms

Our systems-level approach aims to reverse engineer, from mass brain signals, the hierarchy of brain computations that represent and transform sensory inputs to produce behavior—i.e., the brain's algorithm of the behavioral task. The four stages of computation that we systematically found in each

individual participant and tasks meet the five key properties of an algorithm: (1) the inputs were specified as the four possible combinations of two binary inputs; (2) the output responses were also specified as the responses of the logical functions XOR, OR, and AND; (3) the algorithms were definite in each task, with a sequence of two characterized Lin and two NonLin computations that transformed the same inputs into the task-specific outputs; the algorithms were also (4) effective in the brain, in the sense that they only relied on brain resources, and (5) finite in processing time, producing behavior with ~450–500ms.

Note that reverse engineering a systems-level algorithm at the granularity of MEG brain measures does not preclude the existence of different compositions of algorithms at lower levels of granularity of individual neurons that together implement the higher-level algorithm (much like the lower granularity algorithms of machine language implement the higher-level algorithms of C++). Rather, such systems-level analysis provides constraints on where (the brain regions) and when (the specific time windows) specific computations take place, enabling targeted studies of the algorithmic 'how' across modalities and granularities of brain measures. *Jonas and Kording, 2017* used a related systems-level approach to understand the hierarchy of computations of a microprocessor and concluded that there was risk that analytic approaches in neuroimaging could fall short of producing a meaningful algorithmic understanding of neural activity. We could do so here because we adhered to the main properties of an algorithm: our explicit behavioral tasks (i.e. XOR, OR, and AND) require an implementation of a specific computation on simple (i.e. fully characterized) binary inputs to achieve behavior. We could therefore trace the dynamic representations of the inputs into the 2D space of MEG activity to understand the stages of representation underlying the computation (i.e. Lin, LinBoth, and task-specific NonLin). Such descriptive models of an algorithm enable explicit testing of the different stages of the computation hierarchy. For example, by manipulating the timing or nature of the presented stimuli, or by targeting causal interventions (e.g. magnetic stimulation, or stimulus manipulations) at specific brain regions and peristimulus times.

## Generalization to naturalistic categorization tasks

Generalizing from our case study of the algorithms underlying the simple XOR, AND, and OR functions to more naturalistic face, object, and scene categorization tasks will incur many challenges that we can frame in the context of the properties of an algorithm detailed above.

A key challenge is that the task-relevant features of real-world faces, objects, and scenes, may be completely different for different behaviors and participants, effectively changing the inputs to the algorithm. Unfortunately, task- or participant-specific features are generally not considered in studies of neural representation, processing, and categorization. Their understanding remains a similar challenge for deep convolutional neural network research, including instances when these are used as models of the brain. Specifically, a key property of an algorithm is that we specify its inputs as precisely as possible. In real-world categorizations, this implies understanding which specific features of the complex images are task relevant for each particular participant performing each specific behavioral task. Furthermore, specification of the outputs is another key property of an algorithm. Passive viewing, or one-back tasks do not provide this specification. For example, from the same face, the feature of a smiling mouth feature will be used to overtly respond 'happy', but the forehead wrinkles to respond '45 years of age'; from the same car picture, its specific shape to respond 'New Beetle', but the front tyre shape to respond 'flat tyre'; or the specific roof tiles to respond 'Chrysler building' but the density of buildings on the horizon to respond 'city'; and so forth. Relatedly, experts vs. novices will use different features to classify the same pictures of the 35 different subtypes of sparrows that exist in North America. Such relative perceptual expertise and associated features generally characterize the relationship between visual cognition and outside world stimuli. Then, to infer the hierarchical stages of computation from the brain measures, we can start tracing the dynamic representation of these specific task-relevant input features, when we have formally characterized them, between stimulus onset and explicit output task behavior, as we did here. Different modalities or granularities of brain measures (e.g. M/EEG, 3/7T fMRI, NIRS vs. single electrodes (*Gidon et al., 2020*) and electrode arrays) will likely provide complementary understandings (e.g. timing vs. location) of the computations in different brain regions. And when we finally have a model of the computation hierarchy (even initially a descriptive model), we can test its key properties.

To conclude, we reverse engineered four stages of dynamic algorithmic computations over the same sensory inputs that produce different behaviors in the brain. We could do so because we explicitly controlled the input features and the explicit tasks that each individual participant was instructed to resolve while we modeled their brain response with MEG source level activity. Therefore, our results and methodology pave the way to study algorithmic computations when the stimuli and tasks are more complex (e.g. face, object, and scene and their explicit categorizations) but well controlled (e.g. with generative systems rather than uncontrolled 2D images), as they are in any algorithm.

## Materials and methods

### Participants

We recruited 35 participants (all right handed; 24 women). All reported normal or corrected-to-normal vision and gave written informed consent to participate in the experiment and for their data to be anonymously published. We conducted the study according to the British Psychological Society ethics guidelines and was approved by the ethics committee at the College of Medical, Veterinary and Life Sciences, University of Glasgow.

### Stimuli

We synthesized an average face using a generative photorealistic 3D face model (*Yu et al., 2012*; *Zhan et al., 2019a*) to which we added glasses with an image editing program (Adobe Photoshop). Black and clear lenses produced four different input conditions corresponding to four classes of logical inputs: (1) both clear, in vector notation [0,0], (2) left clear/right dark, [0,1]; (3) left dark/right clear [1,0]; and (4) both dark, [1,1]. The edges of the left and the right lens were 0.5 deg of visual angle away from the centrally presented fixation cross.

### Task procedure

Each trial began with a central fixation cross displayed for a randomly chosen duration (between 500 and 1000ms), immediately followed by one of the four stimulus classes described above and displayed for 150ms. We instructed participants to maintain fixation on each trial, to pay attention to the left and the right lenses and to respond as quickly and accurately as possible by pressing one of two keys ascribed to each response choice with the index or middle fingers of their right hand. Responses were 'same' vs. 'different' in the XOR task; 'both dark' vs. 'otherwise' in AND; or 'at least one dark' vs. 'otherwise' in OR. Participants were randomly allocated to one of the three tasks.

Stimuli were presented in blocks of 80 trials, with random intertrial interval of [800–1300ms] and randomized stimulus order in each block. Participants completed a total of 20–24 blocks split across 2–3 single day sessions, with short breaks between blocks. Each session lasted 2.5–3hr. We selected this recording time and number of trials based on sensitivity of MEG for within-participant whole brain corrected feature representation in previous experiments using face stimuli (*Zhan et al., 2019b*; *Ince et al., 2015*). We focus on within-participant inference in which each participant serves as an independent replication of the experiment. Inferring population prevalence of effects is less sensitive to the number of participants and more sensitive to the amount of data collected per participant (*Ince et al., 2021*). Note that the 5–9hr of scanning time per participant we employ ( >200hr scanning in total for the 30 participants in the main experiment) is far higher than typical standards in the field in which N = 30 participants might be scanned for 45min each (total scanning time~22hr).

### MEG data acquisition and preprocessing

We recorded the participants' MEG activity using a 248-magnetometer whole-head system (MAGNES 3600 WH, 4D Neuroimaing) at a 1017 Hz sampling rate. We discarded each participant's runs with more than 0.6 cm head movement measured with prerun vs. postrun head position recordings. Participants were excluded if the number of trials remaining after preprocessing (eye movement artifact rejection and rejecting runs for excessive head motion) was less than 700. We excluded five participants resulting in a final sample sizes of N = 30 (10 per task). Mean head movement (averaged across blocks) across participants was 0.3 cm (min = 0.12, max = 0.44).

We performed analyses with Fieldtrip (*Oostenveld et al., 2011*) in MATLAB, according to recommended guidelines (*Gross et al., 2013*). We high-pass filtered the data at 0.5 Hz (fifth order two-pass

Butterworth IIR filter), filtered for line noise (notch filter in frequency space), and denoised via a Principtal Component Analysis (PCA) projection of the reference channels. We identified noisy channels, jumps, muscle, and other signal artifacts using a combination of automated techniques and visual inspection. The median number of trials for subsequent analyses was 1064 (min = 701, max = 1361).

Next, we epoched the data into trial windows ([−500 to 1000ms]) around stimulus onset, low-pass filtered the data at 45 Hz (third order two-pass Butterworth IIR filter), resampled to 256 Hz, and decomposed using Independent Component Analysis (ICA), separately for each participant. We identified and projected out of the data the ICA sources corresponding to heartbeat and eye blinks or movements (2–4 components per participant).

## Source reconstruction

For each participant, we coregistered their structural MRI scan with their head shape recorded on the first session and warped to standardized MNI coordinate space (*Gross, 2019*). Using brain surfaces segmented from individual warped MRI, we then prepared a realistic single-shell head model. Next, we low-pass filtered the clean dataset at 40 Hz, re-epoched the data between −100 and 400ms around stimulus onset, demeaned using a prestimulus baseline, and computed covariance across the entire epoch. Using average sensor positions across good runs (i.e. where head movement was <0.6 cm, see above), and a 6 mm uniform grid warped to standardized MNI space, we then computed the forward model, keeping the two orientations of MEG activity. We computed the Linearly Constrained Minimum Variance beamformer (*Hillebrand and Barnes, 2005*) solution with parameters 'lambda = 6%' and 'fixedori = no'. The resulting inverse filter applied to the sensor space MEG activity enabled reconstruction of the single-trial 2D MEG magnetic field vector (i.e. dipole with amplitude and direction) activity time courses on 12,773 grid points. Using a Talairaih-Daemon atlas, we excluded all cerebellar and noncortical sources and performed the statistical analyses on the remaining 5107 cortical grid points.

## Linear vs. nonlinear representations

### Linear representation

Every 4ms time point between −100 and 400ms poststimulus, we computed independent multivariate linear regressions to model the dynamic representation of the state of each input (i.e. 0, clear lens vs. 1, dark lens) into the 2D MEG responses of each source. We computed three linear models covering each input separately (Left, L and right, R) and additively.

$$\mathbf{y} = \beta_0 + \beta_1 L$$

$$\mathbf{y} = \beta_0 + \beta_1 R$$

$$\mathbf{y} = \beta_0 + \beta_1 L + \beta_2 R$$

We fitted each model with ordinary least squares, resulting in beta coefficients for the intercept and slope. We quantified the fit in the 2D response space of the source with a multivariate $R^2$ that quantifies multivariate variance as the determinant of the covariance matrix:

$$R^2 = 1 - \frac{\left| (y - \hat{y})^T \, (y - \hat{y}) \right|}{\left| (y - \bar{y})^T \, (y - \bar{y}) \right|}$$

where $y, \bar{y}, \hat{y}$ are the 2D source data, their mean, and model predictions respectively. This linear modeling produced a time course of $R^2$ values per source with 4ms resolution.

To control the FWER over all considered time points and sources, we computed a nonparametric statistical threshold with the method of maximum statistics (*Groppe et al., 2011*). Specifically, on each of 100 permutations we randomly shuffled input state ('on' vs. 'off') across the experimental trials, repeated the linear modeling and $R^2$ computation explained above, and extracted the maximum $R^2$ across all sources and time points. This produced a distribution of 100 maximum $R^2$ values, of which we used the 95th percentile as statistical threshold (FWER p<0.05).

## Nonlinear representation

A fourth model considered, for each source and time point the nonlinear interaction term between the left and Right inputs.

$$y = \beta_0 + \beta_1 L + \beta_2 R$$
$$y = \beta_0 + \beta_1 L + \beta_2 R + \beta_3 L \times R$$

A log-likelihood ratio (LLR) tested whether the added interaction term significantly improved model fit (p<0.05), FDR corrected over time points and sources (*Groppe et al., 2011*; *Benjamini and Yekutieli, 2001*).

## Representation patterns

Linear and nonlinear representations of the two inputs into 2D source activity could form a variety of different patterns. To ensure that these patterns corresponded to expectations (e.g. of an XOR solution), we applied two further computations at each source and time point. First, we computed the pairwise Mahalanobis distances as detailed below between the color-coded 2D distributions of single trial MEG activity in response to each input class (see *Figure 3*). To do so, we averaged the covariance matrices of each pair of input conditions and multiplied the inverse average covariance by the difference of the condition means:

$$dist = \left(\mu 1 - \mu 2\right)^T * C^{-1} * \left(\mu 1 - \mu 2\right)$$

Then, we quantified the geometric relationships between the two-dimensional centroids of the source responses to each input class. We did so by combining the pairwise distances in the way that quantifies the expected representational pattern (see *Figure 4*, right):

- left lens representation (LL): mean([d1, d3, d5, d6]) – mean([d2, d4]). This measure contrasts distances where the left lens state changes, with distances where the left lens state does not change.
- right lens representation (RL): mean([d1, d2, d4, d6]) – mean([d3, d5]). As above, for the right lens.
- both lenses representation (BL): mean(all) – std(all)
- XOR representation: mean([d2, d3, d4, d5]) – mean([d1, d6]). Contrasts the distances between elements of the two different output classes with the distances between elements within each output class.
- AND representation: mean([d4, d3, d6]) – mean([d1, d2, d5])
- OR representation: mean([d6, d2, d5]) – mean([d4, d3, d1])

We tested the statistical significance of each pattern with permutation testing, using again the method of maximum statistics. Over 25 permutations, at each source and time point, we randomly

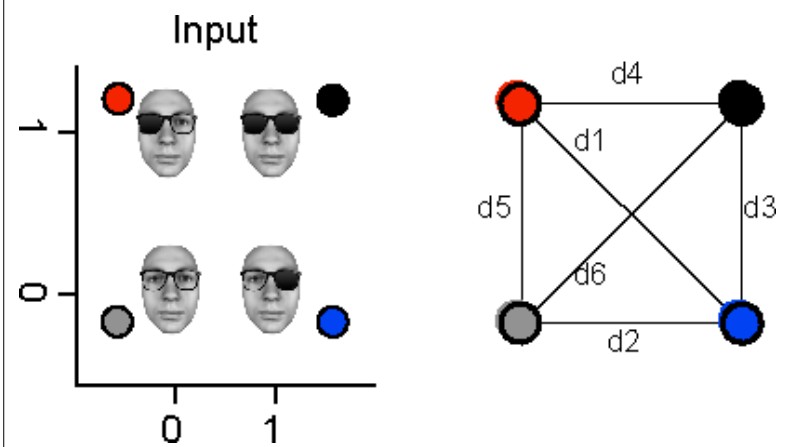

**Figure 4.** Computation of representation patterns. Note: the face stimulus was artificially synthesized and so does not belong to any real person.

shuffled input 'on' and 'off', repeated the above distance calculations, computed the maximum differences, and used the 95th percentiles of these maxima as thresholds (FWER $p<0.05$, corrected).

Finally, we weighted the significant XOR, AND, and OR representation patterns (see above) with the significant $R^2$ for linear representation patterns of the left or right input (see *Methods, Linear Representation*) or with the significant nonlinear LLR test statistic (see *Methods, Nonlinear Representation*).

## Localization of representation patterns

We quantified the temporal dynamics of the four stages of information processing in individual participants as follows. For early representation of the left and right lenses, we computed representation onsets as the first significant time point of $R^2$. For early representation of both lenses, we computed its $R^2$ peak time. Finally, for XOR, AND, and OR nonlinear representations, we also selected the peak times of the respective representation task pattern distance measure. For each state, we then computed time stamps as median across observers and extracted the sourcewise representation averaged across participants at the corresponding time stamp. We plotted these sources back onto glass brains in *Figure 1C* using *Nilearn* (*Abraham et al., 2014*).

## Clustering of the computation stages

To compute the number of computation stages that characterize the whole information processing, we applied a data-driven clustering analysis (k-means) on the 5107 × 102 (source × time points) space separately in XOR, AND, and OR tasks as follows:

### Step 1

First, we transformed each participant's MEG data into the main computation that each source performs at each time point, by assigning to each source the computation (i.e. LinLeft, LinRight, LinBoth, XOR, AND, or OR) with highest representation pattern score at this time point (relative to the participant's distribution across all sources and time points). This produced a source × time matrix of strongest source-level computations for this participant. Examination of the data revealed strong regularities of the computations performed in different time windows (e.g. LinLeft early on), though each computation could be performed across slightly different sources of the same region across participants (e.g. right occipital cortex). Across participants, we therefore computed the modal computation at each source and time point, producing one group-level computation matrix per task that we then clustered over time, rather than over source × time, as we next explain.

### Step 2

In the task-specific computation matrix, we summed at each time point the number of sources that performed each computation (out of six, LinLeft, LinRight, LinBoth, XOR, AND, and OR). The resulting 6 (computations) × 102 time points matrix represented the total brain volume of each computation over time in the task.

### Step 3

We k-means clustered (k = 1–20, repeating 1000 times) each computation matrix from step 2, using the 102 time points as samples and selected *k* as the elbow of the within-cluster sums of point-to-centroid distances metric—i.e., as the furthest point from straight line between k = 1–20. In the XOR, AND, and OR tasks, different clusters therefore represent different stages of the full process over time, with different brain volumes of source-level computations (i.e. LinLeft, LinRight, LinBoth, XOR, AND and OR).

*Figure 1—figure supplement 1A* shows that the XOR, AND, and OR tasks all had k = 5 as a good solution. First, a stage 0, before any computation starts, and then four distinct timed stages with different brain volumes of LinLeft, LinRight, LinBoth, XOR, AND, and OR computations. To visualize each stage, we used the onset time of stages 0, 1, and 2 and the central time point of stages 3 and 4. In each stage, we color coded at voxel level in the small brains the most frequent computation across participants (i.e. LinLeft, LinRight, LinBoth, XOR, AND, and OR).

## Similarity of computation stages across tasks and stimuli

To test whether the XOR, AND, and OR tasks share stages of computation, we computed the percentage of sources that perform the same computations across any pair of stages. This produced

a 12 (3 tasks × 4 stages)-by-12 similarity matrix that compares each stage in each task with all other stages in all other tasks. *Figure 1—figure supplement 1B* reveals that stage 1 (LinLeft and LinRight, cyan and orange) and stage 2 (LinBoth, magenta) are similar across all tasks, whereas stages 3 and 4 (i.e. NonLin, green; NonLin & RT, yellow) are specific to each task.

To test the generalization of computation stages between face and Gabor stimuli, we computed the similarity matrix between the group-level stages in XOR, AND, and OR tasks with faces and the individual participant's (N = 3) stages in XOR with Gabor stimuli (computed as explained with steps 1–3 above, but here within participant). *Figure 1—figure supplement 1B* shows, for each Gabor participant, that their first two linear stages do indeed generalize to the first two linear stages of XOR, AND, and OR faces, whereas their third and fourth NonLinear stages only generalize to the third and fourth stages of XOR faces.

### Information theoretic analyses

We used information theory to quantify the association between RTs and MEG activity, as MI (<RT; $MEG_t$>), splitting RTs into four equiprobable bins and using continuous MEG (on all sources and time points). To this end, we used Gaussian-Copula mutual information *Ince et al., 2017* on all sources and time points. We assessed statistical significance with a permutation test (p<0.05). *Figure 3—figure supplement 1* shows the source × time point average MI and its prevalence across 10 participants in each task.

### Acknowledgements

PGS received support from the Wellcome Trust (Senior Investigator Award, UK; 107802) and the Multidisciplinary University Research Initiative/Engineering and Physical Sciences Research Council (USA, UK; 172046–01). RAAI was supported by the Wellcome Trust [214120/Z/18/Z]. The funders had no role in study design, data collection and analysis, decision to publish or preparation of the manuscript.

## Additional information

### Funding

| Funder | Grant reference number | Author |
| --- | --- | --- |
| Wellcome Trust | | Philippe G Schyns<br>Robin AA Ince |
| Multidisciplinary University Research Initiative | | Philippe G Schyns |
| Wellcome Trust | | Philippe G Schyns<br>Robin AA Ince |

The funders had no role in study design, data collection and interpretation, or the decision to submit the work for publication.

### Author contributions

Katarzyna Jaworska, Conceptualization, Data curation, Formal analysis, Investigation, Methodology, Software, Visualization, Writing – original draft, Writing – review and editing; Yuening Yan, Formal analysis, Writing – original draft, Writing – review and editing; Nicola J van Rijsbergen, Conceptualization, Data curation, Formal analysis, Investigation, Methodology, Software, Supervision, Writing – review and editing; Robin AA Ince, Conceptualization, Formal analysis, Investigation, Methodology, Software, Supervision, Visualization, Writing – review and editing; Philippe G Schyns, Conceptualization, Funding acquisition, Investigation, Methodology, Project administration, Supervision, Visualization, Writing – original draft, Writing – review and editing

### Author ORCIDs

Katarzyna Jaworska http://orcid.org/0000-0001-6482-1498
Yuening Yan http://orcid.org/0000-0003-4027-2687
Robin AA Ince http://orcid.org/0000-0001-8427-0507

Philippe G Schyns ⓘ http://orcid.org/0000-0002-8542-7489

### Ethics

Human subjects: All participants gave written informed consent. We conducted the study according to the British Psychological Society ethics guidelines and was approved by the ethics committee at the College of Medical, Veterinary and Life Sciences, University of Glasgow.

### Decision letter and Author response

Decision letter https://doi.org/10.7554/eLife.73651.sa1
Author response https://doi.org/10.7554/eLife.73651.sa2

---

## Additional files

### Supplementary files
• Transparent reporting form

### Data availability

The analyzed data and custom code that support the findings of this study are deposited in Dryad: https://doi.org/10.5061/dryad.d7wm37q2x. Any further information are available by request to the Lead Contact.

The following dataset was generated:

| Author(s) | Year | Dataset title | Dataset URL | Database and Identifier |
| --- | --- | --- | --- | --- |
| Jaworska K, Yan Y, Nicola JVR, Robin AAI, Philippe GS | 2022 | DifferentComputations_MEG_eLife | https://doi.org/10.5061/dryad.d7wm37q2x | Dryad Digital Repository, 10.5061/dryad.d7wm37q2x |

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
