## [Editor Report]

How does the brain implement basic logical computations (AND, OR, XOR) regardless of stimulus types? This is one of the most fundamental questions in cognitive neuroscience. This MEG study, by combining interesting experimental paradigms and sophisticated signal analyses, demonstrates four serial neural components in different brain regions that correspond to four system-level computations, respectively.

---

## [Decision Letter]

**Decision letter after peer review:**

Thank you for submitting your article "Different computations over the same inputs produce selective behavior in algorithmic brain networks" for consideration by *eLife*. Your article has been reviewed by 2 peer reviewers, one of whom is a member of our Board of Reviewing Editors, and the evaluation has been overseen by Chris Baker as the Senior Editor. The reviewers have opted to remain anonymous.

Essential revisions:

1) Testing the representational generalization across stimulus set. In principle, the logical computation should be independent of the stimuli and would engage similar four-step neural computation in the brain for many types of stimuli. Although the authors demonstrate similar decoding temporal course and patterns for grating and serially presented stimuli (in supplementary materials), it is important to perform a generalization analysis across different stimuli. Specifically, the classifier obtained for face stimuli in the main experiment could be used to decode computations performed over grating stimuli (data in supplementary materials), for example.

2) Clarifying the temporal relationship of the four components and why their alignment over time still supports the four-step conclusion. Moreover, the 3nd component also shows early onset activation. How to reconcile the results with the 4-step computation view? Please see details in the first two comments raised by Reviewer 2.

3) Adding analyses to test alternative computational steps, e.g., 3 system-level, 2 system-level for comparison, and clarifying why the proposed 4-step is the best model to characterize the whole process.

4) The behavioral relevance analysis was only performed on the fourth component. What about the first three components which should in principle be related to behavioral performance as well? More generally, is there any neural signature that is related to behavior for both linear and nonlinear calculations?

5) The authors should collect new data or at least address the possibility of the involvement of eye movement in the four levels of computation.

*Reviewer #1:*

This work by Jaworska et al. examined the spatiotemporal correlates of logical computations over the same stimuli in the human brain using MEG recordings. They revealed four neural components that occur in different brain regions and at varied latencies, corresponding to four system-level computations respectively.

Overall, it is an important study addressing the most fundamental question in cognitive neuroscience, that is, how, where and when do the basic logic computations (AND, OR, XOR) occur in the brain? The study used advanced analysis approaches to address the thorny question and the results are impressively clean and robust and have been replicated for different stimulus sets and conditions.

My major concern is the representational generalization across different stimuli. Specifically, the logical computation (AND, OR, XOR) in principle should be independent of the employed stimuli (face, gratings, etc.) and would thus engage similar neural computation in the brain for different types of stimuli. Therefore, it is important to confirm that the observed logical computation components could be generalized across different stimulus setting (representational generalization across stimuli).

1) If my understanding is correct, the logical computation (AND, OR, XOR) should be independent of the stimuli and therefore would engage similar four-step neural computation in the brain for other types of stimuli. Although the authors demonstrate similar decoding temporal course and patterns for grating and serially presented stimuli (in supplementary materials), I think it is important to perform a generalization analysis across different stimuli. Specifically, we would expect that the classifier obtained for face stimuli could be used to decode computations performed over grating stimuli, for example.

2) I am curious how could the authors make sure that the logical computation (based on the binary classification on four types of stimuli) are the genuine calculation in the brain, since it is completely defined in terms of the experimenter's terminology and the subjects might use different strategy. Have the authors considered other alternative computation over the same inputs? Or put in another way, how could the current results be incorporated into or account for previous findings?

3) The last component is interesting and shows behavioral correlates, which is important evidence. If my understanding is correct, that component only applies to nonlinear integration in combination with RT. What about the behaviorally related linear computation? Is there a general behaviorally related neural component that is related to both linear and nonlinear computation?

*Reviewer #2:*

How our brain dynamically represents and transforms the same visual input into integrated features to produce different behavior remains unclear. To study the dynamic algorithms from mass neural signals, the authors recorded the MEG activity of participants who resolved the classic XOR, OR, and AND behavioral tasks. Using linear and nonlinear representations, they found that source-localized MEG activity progresses through four systems-level computations identified within individual participants. The first two computations similarly represent the inputs across the three tasks; the last two differently represent the inputs in a task-dependent manner. The topic is interesting and timely. The study is elegantly designed and the data statistics are highly significant.

To study the dynamic algorithms from mass neural signals, the authors recorded the MEG activity of participants who resolved the classic XOR, OR, and AND behavioral tasks. Using linear and nonlinear representations, they found that source-localized MEG activity progresses through four systems-level computations identified within individual participants. The first two computations similarly represent the inputs across the three tasks; the last two differently represent the inputs in a task-dependent manner. The topic is interesting and timely. The study is elegantly designed and the data statistics are highly significant. I have some comments listed as below.

1. For each task, the authors proposed the 4 systems-level stages of computation link stimulus to behavior (the first two stages represent and linearly discriminate the visual inputs; the third and fourth stages nonlinearly integrate them in a task-specific manner), according to the different onsets from post-stimulus (~60, 100, 260, and 400ms for each stages, respectively). However, the time window of the first stage (Lin) is the same as the second stage (LinBoth), namely, 74-117ms. This was also reflected very well in the results from Figure 1B, the "Lin" and "LinBoth" computations have almost the same onsets and peaks, particularly, the peak of the blue line (first stage, Lin) looks like much latter than the magenta line (second stage, LinBoth). I am confused that how the authors distinguished these two stages when they had the same time window and why the authors proposed they were the two different stages rather than the same stage during their computations. More generally, the authors should offer more evidence for the 4 systems-level stages of computation, for example, whether it explained their results better than the 3 systems-level stages or 2 systems-level stages of computation.

2. Similar to the first point, for the results in Figure 3, there were several significant peak values in the very early time window (0-60ms) for the third stage ("NonLin"). However, the author only focused on the late time window (~260ms) for this stage. How to reconcile these early peaks (early effects) of the third (late) stage?

3. The behavior data and decoding performance lacked in the first three systems-level computations. Functional inferences based on hypothetical computations of MEG data are not convincing.

4. I would like to suggest the authors collect eye movement data to address whether eye movements are a possible confound for the difference between the four systems-level computations.

5. The legend descriptions are too long and confusing, like figure 2. I would like to suggest the authors label small figures in one figure to make the legend easier to understand by readers.

---

## [Author Response]

Essential revisions:1) Testing the representational generalization across stimulus set. In principle, the logical computation should be independent of the stimuli and would engage similar four-step neural computation in the brain for many types of stimuli. Although the authors demonstrate similar decoding temporal course and patterns for grating and serially presented stimuli (in supplementary materials), it is important to perform a generalization analysis across different stimuli. Specifically, the classifier obtained for face stimuli in the main experiment could be used to decode computations performed over grating stimuli (data in supplementary materials), for example.

We framed this question in the context of the 4-stage clustering analysis by examining the generalization of the Gabor stimuli using the four-stage model derived from the face stimuli. Specifically, we also clustered the Stages of each Gabor XOR participant (N = 3) (see Figure 1—figure supplement 3), leading to four stages in each, and compared each stage within the similarity matrix of the four computation stages over face stimuli (see Figure 1—figure supplement 1). In each Gabor participant, their first two Linear stages are similar to XOR, AND and OR face participants, whereas their third and fourth NonLinear stages are similar only to the third and fourth stages of XOR face participants. This generalizes to Gabor stimuli the 4-stage computation model derived from face stimuli.

2) Clarifying the temporal relationship of the four components and why their alignment over time still supports the four-step conclusion. Moreover, the 3nd component also shows early onset activation. How to reconcile the results with the 4-step computation view? Please see details in the first two comments raised by Reviewer 2.

Clarification of Stages 1 and 2. The alignment in time of the first and second stages in the original manuscript is now clearly separated in each task in the revised figures (see the plots of the revised Figures 1 and 3 in the manuscript reported in a “before” and “after” Author response image 1). The next section explains the revised computations that led to these changes. We know from the anatomy of the visual system that information from a one side of the visual field will first be represented in early visual cortical areas in the contralateral hemisphere. We do indeed find this at Stage 1, but this first stage is very brief (~10-15 ms), because the bottom-up input is quickly propagated across hemispheres to higher visual areas. This duration is consistent with the latencies of cross-hemispheric transfer (~15 ms, Brown et al. 1994; Ipata et al., 1997) also found in our earlier work on cross hemisphere communication of specific visual features (Ince et al. 2016). The second stage that linearly represents both inputs onto the same MEG source in occipital and temporal-ventral cortex is longer (color-coded in purple, see Stage 2 in Figure 1—figure supplement 1 and (Author response image 1)).

**Author response image 1. sa2fig1:** This image shows the changes to the main results presented in Figures 1 and 3 between the original and the revised manuscripts. The color-coded curves present each computation. We can now better see the 4 distinct stages (cf. 4-stage image above), where the brief timing of Stage 1 (in orange and blue, LinLeft and LinRight computations) is now better decoupled from that of Stage 2 (in magenta, LinBoth). Individual participants peak time results for each computation (represented as colored dots) further demonstrates the decoupling into four stages.

Brown WS, Larson EB, Jeeves MA (1994) Directional asymmetries in interhemispheric transmission time: Evidence from visual evoked potentials. *Neuropsychologia* 32:439–448.

Ipata A, Girelli M, Miniussi C, Marzi CA (1997) Interhemispheric transfer of visual information in humans: the role of different callosal channels. *Arch Ital Biol* 135:169–182.

Ince, A.A., Jaworska, K., Gross, J., Panzeri, S., van Rijsbergen, N., Rousselet, G. and Schyns, P.G. (2016). The deceptively simple N170 reflects network information processing mechanisms involving feature coding and transfer across hemispheres. *Cerebral Cortex*, 11, 4123-4135.

Early unexpected peaks. The early peaks of nonlinear computations in Stage 3 (particularly of the OR task) were indeed problematic. We went back to our pipeline, starting with the data of each participant. We noticed in a few participants that an anomaly occurred due to the potential for the task representational distance measure to be noisy, particularly when brain regions had weak representations of the inputs at specific time points. Though the Log-Likelihood Ratio measure of non-linear computation was significant, its effect size was extremely weak compared to the later effects, implying that the task distance was correspondingly less meaningful. We addressed this anomaly by multiplying the statistically significant R2 and LLR (i.e. the effect size for the evidence of the linear and nonlinear relationships) with the statistically significant distance metrics for each computation, to better combine the two measures (and avoid considering distance metrics with very weak effects). This did not change the substance of a 4-stage process but smoothed these anomalies in the plots, as now shown in the key panels of Figures 1 and 3 of the revised manuscript and in Author response image 1. An important side-effect of the better quantifications is the clearer expected separation of Stages 1 and 2 (including in individual participants data), and all stages are sharper in time. We updated all figure and figure supplements accordingly.

3) Adding analyses to test alternative computational steps, e.g., 3 system-level, 2 system-level for comparison, and clarifying why the proposed 4-step is the best model to characterize the whole process.

We thank you and the reviewers for raising these important points. In our methods we described the computations that led to 4 stages, but we did not explicitly test explicitly alternatives numbers of stages (2-, 3-, 4- and more). In our revised manuscript, we now explicitly test these alternative numbers of stages to characterize the whole process. We used a simple data-driven k-means clustering analysis over the measures of each computation across time, pooling all participants’ data and performed separately in the XOR, AND and OR tasks. This created a computation space, where clusters represent different stages of the full process over time, with different brain volumes of source-level computations (i.e. Lin left and right, LinBoth, NonLin XOR, AND and OR). In this space, the data clustered into 4 stages in each task (i.e. with k = 5, determined using the elbow method), with a first null Stage 0 of no computation followed by 4 stages, each characterized with brain volumes of sourcelevel computations (as shown in the new Figure 1—figure supplement 1). Using in each task the indexes to these clusters shows that each cluster does indeed capture a specific stage of the entire process that fits the 4-stage data presented in Figures 1 and 3 of the original paper. To conclude, 4stage is the better model of the entire process.

Furthermore, a similarity analysis of the 4 stages in each task (see Figure 1—figure supplement 1B) revealed the similarity of Stages 1 and 2 of linear computations across the XOR, AND and OR tasks (i.e. Stage 1: Lin left and right, cyan and orange; Stage 2: LinBoth), followed by dissimilar nonlinear computations (i.e. NonLin, green; NonLin and RT) to compute and XOR, AND and OR for task specific behaviors.

4) The behavioral relevance analysis was only performed on the fourth component. What about the first three components which should in principle be related to behavioral performance as well? More generally, is there any neural signature that is related to behavior for both linear and nonlinear calculations?

Thank for you for this comment. We had performed this full analysis but did not report it. It is now included in the manuscript (as Figure 2—figure supplement 1) and we can indeed confirm that the fouth Stage is the only one that relates to behavioral reaction time.

5) The authors should collect new data or at least address the possibility of the involvement of eye movement in the four levels of computation.

This is a difficult point to address for two main reasons: First, we did not collect eye movement data for all participants and so would need to rerun the entire experiment to address this comment. Second, there is no specific proposal for how eye movements could confound the differences between the four systems-level computation stages that we report. Here, we review the arguments that make such confounds unlikely. First, consider that our experimental conditions and task instructions were optimized to minimize eye movement artifacts. Participants were instructed to fixate the center of the screen and not move their eyes throughout the experiment, a fixation cross was presented at the center of the screen on each trial (for a random duration of 500-1000ms), the stimuli were presented for only 150 ms (the timing of only one potential fixation) and preprocessing of our MEG data removed all eye movements artifacts (we now amended the manuscript to make this explicit). Second, further consider that each stage is consistently located in specific brain regions, which are not primarily involved with fixation control (the frontal eye field is). Specifically, we located LinRight and LinLeft at left and right occipital cortex at Stage 1, LinBoth at Stage 2, starting ~15 ms later, in mid-line occipital cortex and right fusiform gyrus, and NonLin XOR, AND and OR task-relevant computations in temporal parietal, parietal and pre-motor cortex. Furthermore, we now show that Stages 1 and 2 are similar across all three tasks, with rapid sequencing (of ~10-15 ms) compatible with the timing of hemispheric transfer (Brown et al., 1994; Ince et al., 2016; Ipata et al., 1997), not with the timing of different fixations (each would take ~150 – 300 ms with face stimuli). The last two NonLin computations are different across the three tasks, but they occur between ~200-400 ms, when the stimulus is not on the screen anymore. In sum, as the two inputs are identical in all tasks, but disappear from the screen (at 150 ms), it is unclear how specific eye movements would confound Stages 3 and 4 (~200-400 ms post-stimulus). These two stages perform task-specific nonlinear integrations (of inputs absent from the screen at this time) over 76 ms in temporal, parietal and premotor cortex (and not in the frontal eye field). In sum, we deem it unlikely that different patterns of eye movement in each task could confound the stages of computation that we report.

Brown WS, Larson EB, Jeeves MA (1994) Directional asymmetries in interhemispheric transmission time: Evidence from visual evoked potentials. *Neuropsychologia* 32:439–448.

Ipata A, Girelli M, Miniussi C, Marzi CA (1997) Interhemispheric transfer of visual information in humans: the role of different callosal channels. *Arch Ital Biol* 135:169–182.

Ince, A.A., Jaworska, K., Gross, J., Panzeri, S., van Rijsbergen, N., Rousselet, G. and Schyns, P.G. (2016). The deceptively simple N170 reflects network information processing mechanisms involving feature coding and transfer across hemispheres. *Cerebral Cortex*, 11, 4123-4135.

Reviewer #1:1) If my understanding is correct, the logical computation (AND, OR, XOR) should be independent of the stimuli and therefore would engage similar four-step neural computation in the brain for other types of stimuli. Although the authors demonstrate similar decoding temporal course and patterns for grating and serially presented stimuli (in supplementary materials), I think it is important to perform a generalization analysis across different stimuli. Specifically, we would expect that the classifier obtained for face stimuli could be used to decode computations performed over grating stimuli, for example.

We thank the reviewer for this comment. Our analyses are mainly within participant and so we do not have an explicit classifier for the group (and such classifier would likely require higher participant numbers). Thus, we addressed the question of generalization by considering explicitly the similarity of stages of computation across the XOR, AND and OR groups of participants we have, with an approach that also speaks to the generalization from face stimuli to other types of stimuli. See the fuller response in point 3 of the response to the Editors and the generalization data in Figure 1—figure supplement 1.

2) I am curious how could the authors make sure that the logical computation (based on the binary classification on four types of stimuli) are the genuine calculation in the brain, since it is completely defined in terms of the experimenter's terminology and the subjects might use different strategy. Have the authors considered other alternative computation over the same inputs? Or put in another way, how could the current results be incorporated into or account for previous findings?

Here, we took the simplest computations that we could think of, documented in theory of computation as a well-known nonlinearity required to resolve XOR (in contrast to AND and OR). Each of these tasks are defined completely by a four element “truth table” and theory of computation consider them to be undecomposable computational primitives. We acknowledge the limitations of such simple tasks, but their advantage is that if the participants are correctly performing it, then we can be confident that somewhere in their brains the XOR computation (i.e. the rule specified in the truth table) is indeed being dynamically implemented. It is difficult to see how different strategies than nonlinear integration could be employed to perform XOR starting with two separate inputs. We did indeed ensure (cf. Stage 1) initial contra-lateral representation of each input (as predicted from neurophysiology and neuroanatomy) to enable a network model that takes in two separate inputs and must necessarily integrate them for behavior. From there on, we traced where and when MEG source activity linearly (Stage 2) and nonlinearly (Stages 3 and 4) represents the same two inputs on the way to behavior.

As we cannot access the ground-truth computations of the brain, all we can do is model them. With each task (XOR, AND and OR) and participant (N = 10 in each group), we now show similar Stages 1 and 2 of linear computations over the two inputs and divergent, task-related Stage 3 and Stage 4 computations over these same inputs (cf. Figure 1—figure supplement 1). Thus, within each task and across the tasks (and now also across stimuli), we are validating that the brain does indeed perform different computations over the same inputs, both at different stages of the process within task and across the final two stages across tasks.

3) The last component is interesting and shows behavioral correlates, which is important evidence. If my understanding is correct, that component only applies to nonlinear integration in combination with RT. What about the behaviorally related linear computation? Is there a general behaviorally related neural component that is related to both linear and nonlinear computation?

Thank for you for this comment. We had performed this analysis but did not report it. We have now included in the manuscript (as Figure 2—figure supplement 1) and can indeed confirm that the fourth component is the only one that relates to behavioral reaction time.

Reviewer #2:The study is elegantly designed and the data statistics are highly significant. I have some comments listed as below.1. For each task, the authors proposed the 4 systems-level stages of computation link stimulus to behavior (the first two stages represent and linearly discriminate the visual inputs; the third and fourth stages nonlinearly integrate them in a task-specific manner), according to the different onsets from post-stimulus (~60, 100, 260, and 400ms for each stages, respectively). However, the time window of the first stage (Lin) is the same as the second stage (LinBoth), namely, 74-117ms. This was also reflected very well in the results from Figure 1B, the "Lin" and "LinBoth" computations have almost the same onsets and peaks, particularly, the peak of the blue line (first stage, Lin) looks like much latter than the magenta line (second stage, LinBoth). I am confused that how the authors distinguished these two stages when they had the same time window and why the authors proposed they were the two different stages rather than the same stage during their computations. More generally, the authors should offer more evidence for the 4 systems-level stages of computation, for example, whether it explained their results better than the 3 systems-level stages or 2 systems-level stages of computation.

We thank the reviewer for this comment, agree with them and have developed a new data-driven analysis that delivers the 4-stage account of the data, rather than fewer or more stages. We presented these results in the context of the overall response to the 5 main points of the Editors.

2. Similar to the first point, for the results in Figure 3, there were several significant peak values in the very early time window (0-60ms) for the third stage ("NonLin"). However, the author only focused on the late time window (~260ms) for this stage. How to reconcile these early peaks (early effects) of the third (late) stage?

We thank again the reviewer for this comment and for pointing to these peaks that caused confusion.

We explained how we addressed the peaks in the response to the Editors and provide a “Before and After” (Author response image 1) that handle these anomalies and revised Figures 1 and 3 accordingly. Note the clearer separation of the early stages (including individual participants data) and their increased sharpness in time.

3. The behavior data and decoding performance lacked in the first three systems-level computations. Functional inferences based on hypothetical computations of MEG data are not convincing.

Thank for you for this comment. We had performed this analysis but did not report it. We have now included in the manuscript (Figure 2—figure supplement) and can indeed confirm that the fourth Stage is the only one that relates to behavioral reaction time. Please see response to the Editors above for the figure showing the data.

4. I would like to suggest the authors collect eye movement data to address whether eye movements are a possible confound for the difference between the four systems-level computations.

See response to this point in response to the Editors’ comments.

5. The legend descriptions are too long and confusing, like figure 2. I would like to suggest the authors label small figures in one figure to make the legend easier to understand by readers.

We agree, have simplified the caption, and added a Supplementary video (Figure 1-video supplement 1) that presents the dynamics that Figure 2 illustrated statically. The figures and new video should make the specifics of the different dynamic computations over the same input a lot easier to understand.